# Is Pontryagin's Maximum Principle All You Need? Solving optimal control problems with PMP-inspired neural networks

## Abstract

Calculus of Variations is the mathematics of functional optimization, i.e., when the solutions are functions over a time interval. This is particularly important when the time interval, or support, is unknown like in minimum-time control problems, so that forward-in-time solutions are not possible. Calculus of Variations also offers a robust framework for learning optimal control and inference with moving boundaries. How can this framework be leveraged to design neural networks to solve challenges in control and inference? We propose the Pontryagin's Maximum Principle Neural Network (PMP-Net) that is tailored to estimate control and inference solutions, in accordance with the necessary conditions outlined by Pontryagin's Maximum Principle. We assess PMP-Net on two classic optimal control and inference problems: optimal linear filtering and minimum-time control. Our findings indicate that PMP-Net can be effectively trained in an unsupervised manner to solve these problems without the need for ground-truth data, successfully deriving the classical "Kalman filter" and "bang-bang" control solution. This establishes a new approach for addressing general, possibly yet unsolved, inference and optimal control problems.

## 1 Introduction

Standard neural networks excel at learning from labeled data, but often lack inherent knowledge of physical principles. In many engineering and scientific applications, there is a wealth of accumulated knowledge and practices that could inform the architecture of learning models. In addition, data in these fields are often scarce, difficult, or expensive to obtain. For instance, telecommunications, data processing, automation, robotics, and control problems frequently have little or no labeled data, making traditional supervised learning methods challenging to apply.

This paper presents a method for designing deep models from first principles by incorporating prior knowledge, specifically existing design principles that have been successful in various engineering, scientific, and technology practices. It focuses on two design problems of broad practical interest: determining the optimal linear estimator and solving the optimal minimum time control problem. We show that our deep models recover the solution to the first, the well-known Kalman filter, and to the second, the bang-bang control, also known as on-off control.

Optimizing over functions with moving boundaries, i.e., when the optimizing variable is a set of whole functions over a variable time interval, falls under the realm of the Calculus of Variations, and a principled solution methodology can be based on Pontryagin's maximum principle (PMP). PMP offers valuable prior knowledge about the necessary conditions for optimal solutions and often provides sufficient conditions, making the optimal solution unique in many cases. This motivates the integration of PMP into machine learning training methodologies.

In this paper, we draw inspiration from the Calculus of Variations—a field focused on finding the maxima and minima of functionals through variations—to design a neural network based on basic principles, which we call the "Pontryagin's Maximum Principle Neural Network" (PMP-Net). This network is designed to solve optimization problems like in Kalman filtering and those arising in control contexts. We start by formulating a variational approach to these problems, using the calculus of variations to derive the necessary conditions for optimization by applying Pontryagin's Maximum

Principle. Although mathematicians and engineers typically solve these conditions analytically or numerically, such methods can be challenging when dealing with nonlinear, second-order differential equations with complex boundary conditions. Instead, we propose using a neural network to learn the optimal solution from PMP's necessary conditions.

Additionally, in minimum-time problems such as the bang-bang control problem, there are two key challenges. First, because the terminal time $t_f$ is to be optimized, the optimization is over a functional space, meaning the optimal solutions are functions over the entire interval $[0, t_f]$, where $t_f$ itself is unknown. As a result, the forward method can not be used and the performance metrics are only valid for admissible trajectories (the trajectory that reaches the final state. Second, the extra constraints, such as functions being bounded or living in a compact set, restrict the control functions to be learned. The optimal solution in these cases is often discontinuous, resembling a step function, and may be undefined in certain regions. These complexities frequently result in vanishing or exploding gradients during neural network training and no prior work overcome these challenges.

This paper presents a method to integrate prior knowledge from Calculus of Variations, functional optimization, and classical control into the architectural design of deep models. We incorporate dynamical constraints, control constraints, and conditions derived from PMP into the loss function for training neural networks, enabling unsupervised learning. Our contributions are as follows.

**Main contributions**:

- Incorporate calculus of variations and Pontryagin's Maximum Principle as soft constraints in ML training methodology and minimizing optimality conditions residual instead of minimizing actual performance metrics. This provides a benefit when the performance functional cannot always be computed.
- Engineer a novel neural network architecture, PMP-Net, that mimics the design of feedback controllers used in optimal control. This allows PMP-Net to apply to different time horizons.
- Propose learning paradigms that effectively train PMP-Net to derive the optimal solution.
- Show that our PMP-Net replicates the design of the Kalman filter and the bang-bang control without using labeled data.

## 2 THEORY

### 2.1 OPTIMAL CONTROL PROBLEM

We illustrate our approach in the context of a control problem. Given an initial value problem, specified by a dynamical system and its initial condition

$$\dot{x}(t) = f(x(t), u(t))$$
$$x(0) = x_0 \tag{1}$$

where $x : \mathbb{R}_{\geq 0} \mapsto \mathcal{X} \subseteq \mathbb{R}^m$ is the state function, $u : \mathbb{R}_{\geq 0} \mapsto \mathcal{U} \subseteq \mathbb{R}^n$ is the control function, and $f : \mathcal{X} \times \mathcal{U} \mapsto \mathcal{X}$ is a known function representing the dynamics. We suppose that $x$ is differentiable and $f$ is differentiable with respect to each variable. Unlike previous works that consider fixed support (Mowlavi & Nabi, 2023) or fixed terminal state (D'Ambrosio et al., 2021), we consider a more general stopping set $\mathcal{S} = \{(x(t), t) | s(x(t), t) = 0\} = \mathcal{X} \times \mathbb{R}_{\geq 0}$ where $s : \mathbb{R}^m \times \mathbb{R}_{\geq \mathbf{0}} \mapsto \mathbb{R}^k$ is differentiable with respect to each variable. This definition of $\mathcal{S}$ allows us to solve general optimal control problems when the terminal state and time are not explicitly specified, e.g., finding the distance between two curves or finding the minimum time to reach the surface of a manifold. In these cases, we do not know the terminal point and terminal time beforehand.

Optimal control problems involve finding for example a control function $u^\star : [0, t_f] \mapsto \mathcal{U}$ such that the corresponding trajectory $(x^\star(t), u^\star(t))_{t \in [0, t_f]}$ reaches the terminal value $(x^\star(t_f), t_f) \in \mathcal{S}$ and minimizes some performance measure $J(x, u)$ of the form

$$J(x, u) = q_T(x(t_f), t_f) + \int_0^{t_f} g(x(t), u(t))dt \tag{2}$$

where $q_T$ is the terminal cost and $g$ is the running cost. Not all pairs of functions $(x, u)$ are admissible trajectories since trajectories must satisfy a dynamical constraint $\dot{x}(t) = f(x(t), u(t))$ and $(x(t_f), t_f) \in \mathcal{S}$. The domain of integration $[0, t_f]$ can be variable, depending on each admissible control. The optimal control problem is therefore the constrained optimization

$$
\begin{aligned}
\min_{x,u} \quad & J(x, u) \\
\text{s.t.} \quad & \dot{x}(t) = f(x(t), u(t)), \forall t \in [0, t_f] \\
& x(0) = x_0, (x(t_f), t_f) \in \mathcal{S}
\end{aligned}
\tag{3}
$$

In equation 3, the optimization variables are functions over variable support, say $\{u(t), t \in [0, t_f]\}$, where $t_f$ may be fixed or is to be optimized itself (like in the minimum time problem).

To handle dynamics constraints, the (function vector) Langragian multipliers $\lambda(t)$ is introduced and the new performance measure becomes

$$
\mathcal{L}(x, u, \lambda) = q_T(x(t_f), t_f) + \int_0^{t_f} g(x(t), u(t)) + \lambda(t)^T (f(x(t), u(t)) - \dot{x}(t)) dt
\tag{4}
$$

For all admissible trajectories $(x, u)$, we have $\mathcal{L}(x, u, \lambda) = J(x, u)$. Therefore, the admissible optimal solution for equation 4 is also the optimal solution for equation 3.

Calculus of variations enables us to identify the optimal functions $(x, u, \lambda)$ that minimize $\mathcal{L}$. By examining variations, we can derive the necessary conditions — known as Pontryagin's maximum principle (PMP) — at the optimal solution $(x^\star, u^\star, \lambda^\star)$ for equation 4.

$$
\begin{aligned}
\dot{x}^\star &= f(x^\star, u^\star) \\
\dot{\lambda}^{\star T} &= -\frac{\partial \mathcal{H}}{\partial x}\Big|_\star \\
u^\star &= \arg\min_u \mathcal{H}(x^\star, u, \lambda^\star) \\
x^\star(0) &= x_0 \\
s(x^\star(t_f), t_f) &= \mathbf{0}
\end{aligned}
\tag{5}
$$

$$
\frac{\partial q_T}{\partial x}(x^\star(t_f), t_f) - \lambda^\star(t_f) = \sum_{i=1}^k d_i \left[ \frac{\partial s_i}{\partial x}(x^\star(t_f), t_f) \right] \quad (\mathcal{B}_1)
$$

$$
\mathcal{H}(x^\star(t_f), u^\star(t_f), \lambda^\star(t_f)) + \frac{\partial q_T}{\partial t}(x^\star(t_f), t_f) = \sum_{i=1}^k d_i \left[ \frac{\partial s_i}{\partial t}(x^\star(t_f), t_f) \right] \quad (\mathcal{B}_2)
$$

where $\mathcal{H}$ denotes the scalar function called the "Hamiltonian," defined as $\mathcal{H}(x(t), u(t), \lambda(t)) = g(x(t), u(t)) + \lambda(t)^T f(x(t), u(t))$. The variables $d_1, ..., d_k$ are to be learned and enforce the terminal state to be in a general stopping set $S$. The system of partial differential equation 5 is generally nonlinear, time-varying, second-order, and hard-to-solve. Numerical methods also pose challenges due to the split boundary conditions–neither the initial values $(x(0), \dot{x}(0))$ nor the final values $(\lambda(t_f), \dot{\lambda}(t_f))$ are fully known.

## 2.2 Pontryagin's Maximum Principle Network

Instead of solving equation 5 analytically or numerically, we propose leveraging neural networks' well-known capability as universal function approximators (Cybenko, 1989) to learn $\{x(t), u(t), \lambda(t), t \in [0, t_f]\}$, along with the learnable parameters $\{t_f, d_1, ..., d_k\}$, that satisfy PMP. In the training stage, rather than directly matching the PMP-Net's outputs to ground truth data $\{x(t)^\star, u(t)^\star, \lambda(t)^\star, t \in [0, t_f]\}$, our PMP-Net **learns** to predict solutions that adhere to the PMP constraints. Because this process incorporates a solution methodology, the PMP, we interpret it as bringing to the neural networks "prior knowledge"(Betti & Gori, 2016). Our approach introduces an inductive bias into the PMP-Net, allowing it to learn the optimal solution in an unsupervised manner. By simultaneously predicting both the state and the control, PMP-Net eliminates the need for integration and can address optimal control problems with unknown terminal time.

During the forward pass, PMP-Net takes time as input and predicts the state $x(t)$, the control $u(t)$, and the costate $\lambda(t)$. The Hamiltonian $\mathcal{H}$ is then calculated based on these predictions. By leveraging the automatic differentiation capabilities of neural networks (Baydin et al., 2017), we can efficiently compute the derivatives and partial derivatives present in equation 5 by computing in-graph gradients of the relevant output nodes with respect to their corresponding inputs. We calculate the residuals of the differential equations in PMP and incorporate them into the loss function, along with the $L_2$ loss between the predicted and target states at the boundary conditions. In the experiments in Sections 3 and 4, we also incorporate additional architectural features into our PMP-Net to enforce hard constraints and to allow PMP-Net to learn even when the terminal time is unknown.

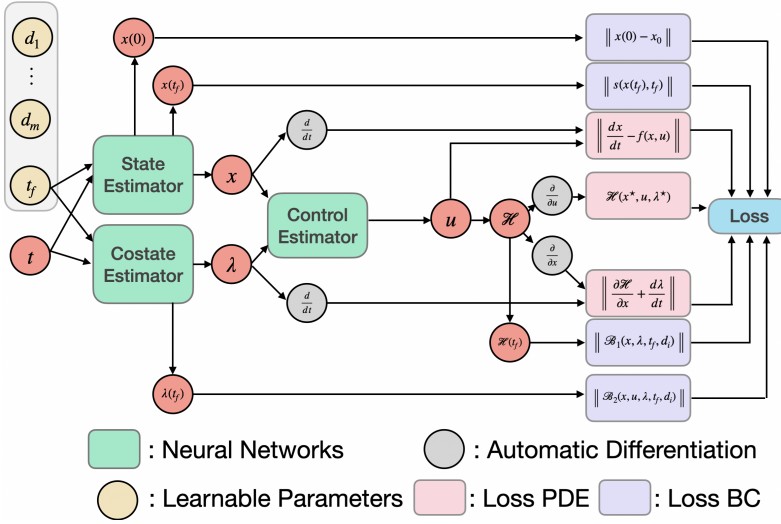

Figure 1: PMP-Net architecture. The state estimator, the costate estimator, and the control estimator are neural networks. We compute the Hamiltonian $\mathcal{H}$, relevant derivatives, and residual of differential equations in PMP. The total loss function consists of loss from residuals and loss from boundary conditions. The variables $t_f, d_1, ..., d_m$ are learnable parameters. Since all loss terms are calculated based on predictions, no labeled data is needed for training

## 3 DESIGNING THE OPTIMAL LINEAR FILTER

In this section, our goal is to design a linear filter that provides the best estimate of the current state based on noisy observations. The optimal solution is known as "Kalman Filtering" (Kalman & Bucy, 1961), which is one of the most practical and computationally efficient methods for solving estimation, tracking, and prediction problems. The Kalman filter has been widely applied in various fields from satellite data assimilation in physical oceanography, to econometric studies, or to aerospace-related challenges (Leonard et al., 1985; Auger et al., 2013). The optimal solution being known, the Kalman filter is the ground truth that serves to benchmark PMP-Net.

### 3.1 KALMAN FILTER

Reference Athans & Tse (1967) formulated a variational approach to derive the Kalman filter as an optimal control problem. We consider the dynamical system

$$
\begin{aligned}
\dot{x}(t) &= Ax(t) + Bw(t), \ 0 \leq t \leq t_f, \quad w_{t-1} \sim \mathcal{N}(0, \mathbf{Q}) \\
y(t) &= Cx(t) + v(t), \ v_{t-1} \sim \mathcal{N}(0, \mathbf{R}) \\
x(0) &\sim \mathcal{N}(x_0, \Sigma_0)
\end{aligned}
\tag{6}
$$

where $x(t) \in \mathbb{R}^n$ is the state, $y(t) \in \mathbb{R}^m$ is the observation. $A \in \mathbb{R}^{n \times n}$ is the state transition matrix, $B \in \mathbb{R}^{n \times r}$ is the input matrix, and $C \in \mathbb{R}^{n \times r}$ is the measurement matrix. The white Gaussian noise $w(t)$ (resp. $v(t)$) is the process (resp. measurement) with covariance $\mathbf{Q}$ (resp. $\mathbf{R}$) noise. We

assume that $x(0), w(t), v(t)$, are independent of each other. Kalman designed a recursive filter that estimates the state by

$$\dot{\hat{x}}(t) = A\hat{x}(t) + G(t)\Big[Cy(t) - A\hat{x}(t)\Big]$$
$$\hat{x}(0) = x_0 \tag{7}$$

where $G(t)$ is the Kalman gain to be determined. Given the state estimation $\hat{x}(t)$ at time $t$, the error covariance defined as

$$\Sigma(t) = \mathbb{E}\Big[(\hat{x} - x)(\hat{x} - x)^T\Big]$$

has the following dynamics

$$\dot{\Sigma}(t) = \Big[A - G(t)C\Big]\Sigma(t) + \Sigma(t)\Big[A - G(t)C\Big]^T$$
$$+ BQB^T + G(t)RG(t)^T \tag{8}$$
$$\Sigma(0) = \Sigma_0$$

where $\Sigma(t)$ is the $n \times n$ error covariance matrix. The goal of Kalman filter is to find the optimal gain (perceived in this variational approach as a control) $G^\star(t)$ such that the final cost

$$q_T(\Sigma(T)) = \text{tr}\left[\Sigma(T)\right]$$

is minimized, or equivalently the $L_2$ norm between the estimation and the actual state is minimized. In this case, the stopping set is $\mathcal{S} = \{(\Sigma(t), t) \mid t = T\}$. Applying Pontryagin's maximum principle to equation 8 (see Appendix B), the necessary conditions to solve for the optimal Kalman gain are

$$\dot{\Sigma}^\star = f(\Sigma^\star, G^\star)$$
$$\dot{\lambda^\star}^T = -\frac{\partial \mathcal{H}}{\partial \Sigma}\Big|_\star$$
$$\frac{\partial \mathcal{H}}{\partial G}\Big|_\star = 0 \tag{9}$$
$$\lambda^\star(T)^T = \boldsymbol{I}_n$$

where the Hamiltonian $\mathcal{H} = \text{tr}\left[\lambda^T f(\Sigma^\star, G^\star)\right]$

## 3.2 LEARNING THE KALMAN FILTER WITH PMP-NET

**Architecture**: The PMP-Net architecture follows the architecture shown in Figure 1. Since $\Sigma$ is both symmetric and positive semi-definite, we embed this inductive bias into our neural network architecture. Specifically, the state estimator outputs an intermediate matrix $P$ and estimates the error covariance $\Sigma$ as $\Sigma = P^T P$, ensuring symmetry and positive semi-definiteness. We adopt the feedback loop design in engineering so that the control estimator only takes the output state as input. The state, costate, and control estimators are modeled by 6-layer feedforward neural networks with hyperbolic tangent activation.

**Training**: We adopt curriculum training, as optimizing loss with multiple soft constraints can be challenging (Krishnapriyan et al., 2021). We set the loss function to be

$$\text{Loss}_\theta = \text{Loss}_{\text{BC}} + \alpha\text{Loss}_{\text{PDE}}$$

where

$$\text{Loss}_{\text{BC}} = \|\Sigma(0) - \Sigma_0\|_2 + \|\lambda(T) - I_n\|_2$$

$$\text{Loss}_{\text{PDE}} = \frac{1}{N}\sum_{i=0}^{N}\|\dot{\Sigma}(t_i) - f(\Sigma(t_i), G(t_i))\|_2 + \left\|\dot{\lambda}(t_i) + \frac{\partial \mathcal{H}(\Sigma, G, \lambda, t_i)}{\partial \Sigma}\right\|_2 + \left\|\frac{\partial \mathcal{H}}{\partial G}\Big|_{t_i}\right\|_2$$

During each epoch, 5000 points are uniformly sampled from time $[0, T]$. After every 5000 epochs, we increment the value of $\alpha$ by a factor of $1.04$. All neural networks are initialized with Glorot uniform initialization (Glorot & Bengio, 2010). We train PMP-Net using stochastic gradient descent with the initial learning rate $8 \times 10^{-4}$.

**Evaluation**: For a fair evaluation, we take the estimated control from PMP-Net and use the fourth-order Runge-Kutta integrator (Runge, 1895) in `scipy.integrate.solve_ivp` to derive the trajectory of the state. This is necessary because the state estimated by PMP-Net might not adhere to the dynamics constraints, making it into an implausible trajectory.

### 3.3 RESULTS

For our experiment, we set

$$A = \begin{bmatrix} \mathbf{0} & \mathbf{I}_2 \\ \mathbf{0} & \mathbf{0} \end{bmatrix} \in \mathbb{R}^{4 \times 4}, B = \begin{bmatrix} \mathbf{0} \\ \mathbf{I}_2 \end{bmatrix} \in \mathbb{R}^{4 \times 2}, C = \mathbf{I}_4, Q = 0.5 \mathbf{I}_2, R = \begin{bmatrix} 4.0 & 1.5 & 0 & 0 \\ 1.5 & 4.0 & 0 & 0 \\ 0 & 0 & 2.0 & 1.0 \\ 0 & 0 & 1.0 & 2.0 \end{bmatrix}, T = 5.0$$

This dynamical system models a kinematics system where the state $x$ corresponds to position and velocity and the control $u$ corresponds to the force applied to the state. With these experimental settings, Kalman filtering reaches a steady state where $\Sigma^\star$ converges (hence, the Kalman gain converges to $G_\infty^\star$). We compare our method against two baselines: 1) the baseline NN trained with 50 points of ground truth control $G^\star$ sampled from the time interval $[0, 2.0]$, covering the transient phase of the Kalman filter before it reaches steady-state and, 2) PINN that enforces the dynamics constraints and minimize the cost functional $q_T$ (Mowlavi & Nabi, 2023). We evaluate and compare the trace of

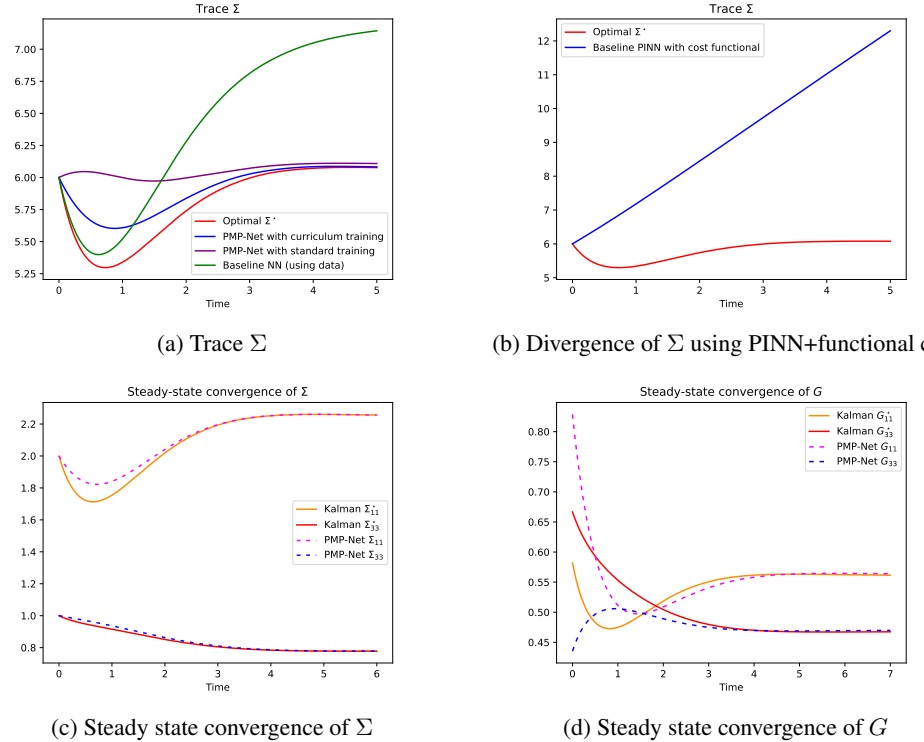

(a) Trace $\Sigma$

(b) Divergence of $\Sigma$ using PINN+functional cost

(c) Steady state convergence of $\Sigma$

(d) Steady state convergence of $G$

Figure 2: PMP-Net learns the Kalman filter, deriving the optimal value of the functional cost. The baseline NN performs well in the time interval where ground truth is available, it fails to learn the optimal steady-state Kalman gain $G_\infty^\star$, resulting in diverging error. The baseline PINN shows diverging error. PMP-Net learns the optimal steady-state error covariance $\Sigma_\infty^\star$ and Kalman gain $G_\infty^\star$ and they remain convergent beyond the time interval of the problem $[0, 5]$

$\Sigma$ generated by PMP-Net, the baseline methods, and the optimal Kalman gain. Since the objective is to minimize the trace of $\Sigma$ at the terminal time $T$, we focus primarily on the final value $\text{tr}(\Sigma(T))$. Figure 2a shows that, even though there is some discrepancy between the PMP-Net's control output $G$ and the Kalman gain $G^\star$ during the transient phase, PMP-Net matches the optimal Kalman gain $G_\infty^\star$ at the terminal time, while the baseline diverges. Figure 2b shows that the baseline PINN that learns to satisfy dynamics constraint and to minimize the cost functional without using optimality conditions shows a divergent behavior. This result demonstrates that including all optimality conditions does not necessarily make optimizing neural networks harder. Figure 2c and 2d show that the PMP-Net's trajectory of $(\Sigma, G)$ converges to their corresponding optimal values $(\Sigma_\infty^\star, G_\infty^\star)$. Since

the gain $G$ is learned as a function of one input $\Sigma$, $(\Sigma, G)$ remains convergent even after time interval of the problem $[0, 5]$, allowing us to use PMP-Net in different time horizon. One can say that PMP-Net learns the correct relationship between $\Sigma_\infty^\star$ and $G_\infty^\star$, equivalent to deriving the Riccati equation. Furthermore, running the optimal filter gain, the error covariance, and the loss $\text{tr}(\Sigma)$ beyond time $T = 5$, they all remain close to the ground truth of the analytical solution. The discrepancy during the transient time does not affect the overall performance, since PMP-Net's control $G$ converges to the optimal steady-state value $G_\infty^\star$. In practice, this is what usually matters, since in Kalman filter practice, the steady-state $G_\infty^\star$ is often pre-computed and used instead of $G^\star(t)$.

We investigated the effect of using curriculum training. As shown in Fig 2a, using curriculum training results in a trajectory with a smaller trace of the error covariance throughout the interval of interest, especially during the transient phase. We leave as future work the optimization of $G$ during the transient phase.

## 4 LEARNING THE MINIMUM TIME OPTIMAL CONTROL

In this section, we seek the optimal control strategy that drives a state from an arbitrary initial position to a specified terminal position in the shortest possible time. In practice, the control is subject to constraints, such as maximum output levels. The optimal control strategy for the minimum time problem is commonly known as "bang-bang" control. Examples of bang-bang control applications include guiding a rocket to the moon in the shortest time possible while adhering to acceleration constraints (Athans & Falb, 1996).

### 4.1 THE MINIMUM TIME PROBLEM

We illustrate the PMP-Net with the following problem. Consider the kinematics system

$$\begin{bmatrix} \dot{x}_1(t) \\ \dot{x}_2(t) \end{bmatrix} = \begin{bmatrix} 0 & 1 \\ 0 & 0 \end{bmatrix} \begin{bmatrix} x_1(t) \\ x_2(t) \end{bmatrix} + \begin{bmatrix} 0 \\ 1 \end{bmatrix} u$$

where $x_1, x_2, u$ correspond to the position, velocity, and acceleration of a mobile platform. The goal is to drive the system from the initial state $(x_1(0), x_2(0)) = (p_0, v_0)$ to a final destination $(p_f, v_f)$ where $x(t) \in \mathbb{R}^n$ is the state at time $t$, $u(t) \in \mathbb{R}^m$ is the control at time $t$. We are interested in finding the optimal control $\left\{ u^\star(t), t \in [0, t_f^\star] \right\}$ that drives the state from $x_0$ to $x_f$ in a minimum time $t_f^\star$. The performance measure can be written as

$$J(x, u) = \int_0^{t_f} 1 dt \tag{10}$$

where $t_f$ is the time in which the sequence $(x, u)$ reaches the terminal state. Note that here $t_f$ is a function of $(x, u)$ since the time to reach the target state depends on the state and control. In practice, the control components may be constrained by requirements such as a maximum acceleration or maximum thrust

$$|u_i(t)| \leq 1, \quad i \in [1, m] \quad t \in [t_0, t_f] \tag{11}$$

where $u_i$ is the $i$th component of $u$. The stopping set for the minimum time problem is $\mathcal{S} = \{(x(t), t) \mid x(t) = \mathbf{0}\}$. Pontryagin's maximum principle gives us the necessary conditions at the optimal solution $(x^\star, u^\star, \lambda^\star)$ for equation 12.

$$\begin{bmatrix} \dot{x}_1^\star \\ \dot{x}_2^\star \end{bmatrix} = \begin{bmatrix} x_1^\star \\ 0 \end{bmatrix} + \begin{bmatrix} 0 \\ u^\star \end{bmatrix}$$

$$\dot{\lambda}^\star = \begin{bmatrix} 0 \\ -\lambda_1^\star \end{bmatrix}$$

$$u^\star = \arg\min_u 1 + \lambda_1^\star x_2^\star + \lambda_2^\star u \tag{12}$$

$$x(t_f^\star) = \mathbf{0}$$

$$\lambda_2(t_f^\star) = - \begin{bmatrix} d_1 \\ d_2 \end{bmatrix}$$

$$1 + \lambda_2(t_f^\star) u(t_f^\star) = 0$$

Since the variables $d_1, d_2$ only appear in one equation, they become redundant. The only additional parameter is $t_f$

## 4.2 LEARNING BANG-BANG CONTROL WITH PMP-NET

**Architecture**: PMP-Net for the minimum time problem is followed from Figure 1. In our approach, the state estimator, costate estimator, and control estimator are modeled by 6-layer feedforward networks. The learnable parameter $t_f$ is subjected to the constraint $x(t_f) = x_f$. Since $d_1, d_2$ are redundant, they are removed from the training.

**Training**: We propose a new paradigm for training PMP-Net for minimum-time problems. First, we set a time $T$ that is sufficiently larger than $t_f^\star$. We start by pretraining the costate estimator such that the costate estimator is not a zero function (see Appendix C). This can be achieved by training the costate estimator to output $a$ at time 0 and $b$ at time $T$, where $a, b$ are heuristic non-zero values. Secondly, we propose sequential and alternate training. The equation 12 suggests that the optimal control $u^\star$ as a function of $(x^\star, \lambda^\star)$ can be learned without knowing $(x^\star, \lambda^\star)$. Therefore, in the first step, we can generate a random $(x, \lambda)$ and train the control estimator to minimize $\mathcal{H}(x, \lambda, u)$. We freeze the state and costate estimator and take $n$ gradient update for the control estimator since $u^\star = \arg\min_u \mathcal{H}(x, \lambda, u)$. Next, we freeze the control estimator and train the state and costate estimator by uniformly sampling 5000 points from time interval $[0, T]$ and perform one gradient update for the state and costate estimator before going back to the first step again. This can prevent vanishing gradients or exploding gradients. We also compute the gradient of the loss function with respect to the variable $t_f$, allowing it to be optimized during backpropagation. . We train PMP-Net using stochastic gradient descent with the initial learning rate $8 \times 10^{-4}$.

**Evaluation**: Similar to the experiment in Section 3.2, we generate the control estimate from PMP-Net and use a fourth-order Runge-Kutta integrator to estimate the state trajectory. For the baseline, we employ the optimal (bang-bang) control and integrate it with the fourth-order Runge-Kutta method. During prediction, we consider the state to have reached the target if the Euclidean distance between them is less than $\epsilon = 0.05$.

## 4.3 RESULTS

For our experiment, we set $x_0 = \begin{bmatrix} 1 \\ 0 \end{bmatrix}, x_f = \begin{bmatrix} 0 \\ 0 \end{bmatrix}, T = 3.0$. The optimal control is to apply the acceleration $-1$ from time $[0, 1]$ and acceleration $+1$ from time $]1, 2]$ that will drive the state from the initial state $x_0$ to the target state $x_f$ in minimum time $t_f^\star = 2$ seconds. The control switches from $-1$ to $+1$ at the switching time at $t = 1$ where $\lambda_2^\star(t) = 0$ as shown in Fig 3b

Figures 3a and 3b show that the generated trajectory of state and costate match the optimal solution. Figure 3c shows that PMP-Net learns a control strategy that exhibits "bang-bang" behavior, switching from $+1$ to $-1$ when $\lambda_2$ changes sign. Since standard neural networks inherently produce continuous functions, there is a small discrepancy between the predicted control and the theoretical bang-bang control, as shown in Fig 3c. This limitation may, in fact, better reflect real-world scenarios, as the control cannot switch instantaneously between two extremes. While reducing this discrepancy is possible by using a larger control estimator and more computational resources to compute gradients of higher magnitude, such optimization is beyond the scope of this work. Figure 3d demonstrates that the trainable variable $t_f$ in PMP-Net successfully converges to the true value of $t_f^\star = 2$. This key result highlights PMP-Net's ability to learn when the terminal time is unknown.

We also conducted an ablation study to examine the impact of our training methods. Fig 4a demonstrates that when the costate estimator is initialized near the zero function, PMP-Net struggles to learn effectively, resulting in loss divergence. Moreover, we investigated the effect of adding the generated $x$ and $\lambda$ to train the control estimator. Fig 4b shows that the output control by the control estimator trained without using the generated $(x, \lambda)$ does not switch at $\lambda_2 = 0$ and its rate of switching between two extremes is gentler.

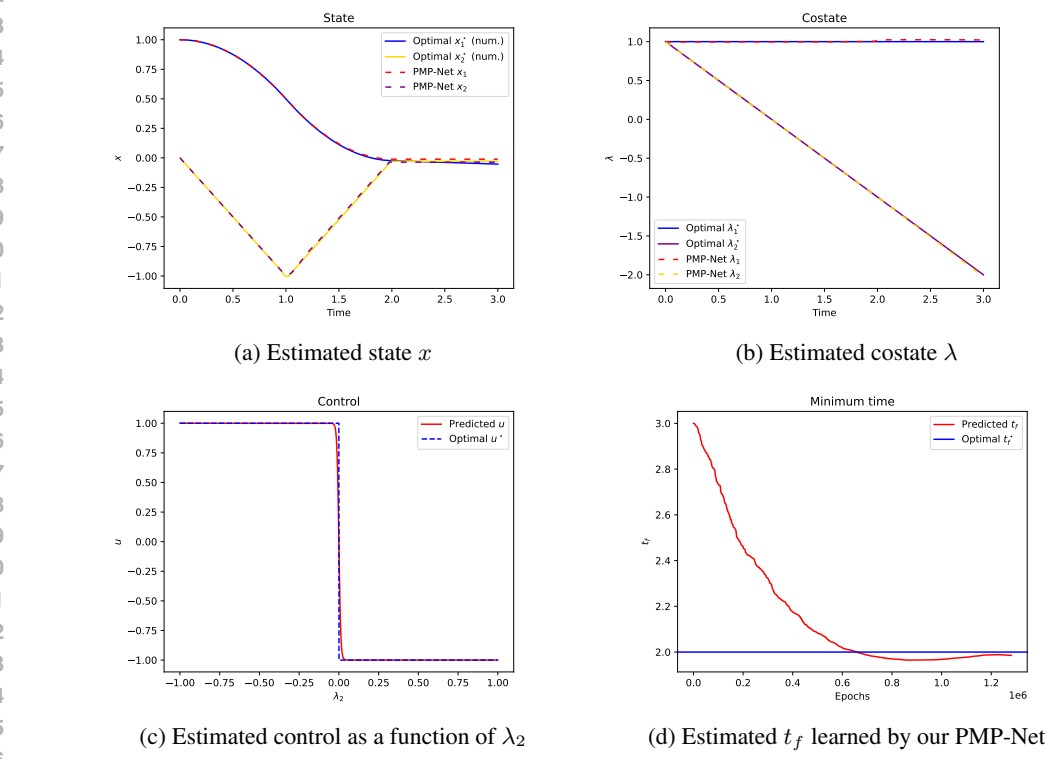

(a) Estimated state $x$

(b) Estimated costate $\lambda$

(c) Estimated control as a function of $\lambda_2$

(d) Estimated $t_f$ learned by our PMP-Net

Figure 3: Learning the optimal control for the minimum time problem with PMP-Net. PMP-Net generates the trajectory of the state, the costate, and the control over the time interval of interest that matches the optimal trajectory. Most importantly, PMP-Net learns the bang-bang behavior where control $u$ is a negative sign function of $\lambda_2$ and correctly learns the minimum time $t_f^\star$.

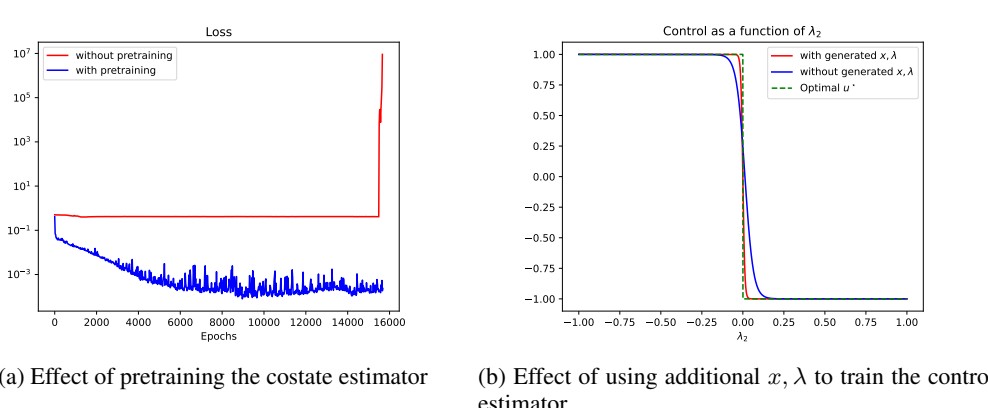

(a) Effect of pretraining the costate estimator

(b) Effect of using additional $x, \lambda$ to train the control estimator

Figure 4: Ablation study. We investigate the effect of pretraining the costate estimator and the effect of using additional $x, \lambda$ to train the control estimator. Fig 4a): PMP-Net fails to train when the costate estimator is not pretrained and is initialized close to zero. Fig 4b): Generating additional $x, \lambda$ data to train the control estimator reduces the discrepancy between the learned control function and the optimal bang-bang solution.

## 5 RELATED WORK

Our approach aligns with the use of neural networks for solving optimal control problems and is inspired by existing literature on integrating constraints into neural network architectures. Below, we provide a concise overview of these areas, highlighting their relevance. We provide a brief overview of these areas and emphasize how our work distinguishes itself from them.

**Enforcing dynamics constraints in neural networks**: Dynamics constraints in neural networks can be addressed through two main approaches: (1) designing specialized architectures that inherently satisfy the constraints (hard constraints), and (2) incorporating the constraints into the loss function, as done in Physics-Informed Neural Networks (PINNs) Raissi et al. (2019) (soft constraints).

In hard constraint approaches, Böttcher et al. (2022) enforce dynamic constraints using neural ODEs (Chen et al., 2018) to learn the optimal control. ODE-based methods primarily address the forward problem by integrating the state to the terminal time, calculating the loss function, and minimizing it. This framework is not applicable when the terminal time $t_f$ is unknown and must be optimized, or when the terminal state is prescribed. Similarly, D'Ambrosio et al. (2021) parameterize the state $x$ and express the control $u$ in terms of $x$ and its higher-order derivatives to satisfy the dynamic constraints. However, such a representation is not always feasible in general dynamics.

In a soft constraint approach, Mowlavi & Nabi (2023) employ PINNs to parameterize the state $x$ and control $u$, ensuring they satisfy the dynamics. The neural network weights are then updated to minimize the performance metric. However, this direct method assumes the performance metric can always be calculated—requiring the supports of the relevant functions to be fixed and known.

In contrast, our method uses the indirect method by leveraging the calculus of variations, enabling us to address cases where the terminal time and terminal state are variables (moving boundary). Our approach simultaneously learns the optimal control and the minimum time, even under these conditions.

**Incorporating optimality conditions in neural networks**: Several works have used optimality conditions of constrained optimization in neural networks. Reference Amos & Kolter (2017) and Donti et al. (2021) incorporate Karush–Kuhn–Tucker (KKT) conditions in implementing backward passes in neural networks. But this is constrained optimization over constant variables (parameters) while we optimize over functions with a dynamic constraint. Reference Yin et al. (2024) and Betti et al. (2024) propose using neural networks to parameterize the state and costate that learns to satisfy KKT and PMP conditions. However, these works only consider problems where the support is fixed. This approach can not be extended to a problem where the support is unknown, e.g., as in the minimum time problem. While D'Ambrosio et al. (2021) considers learning the terminal time, their approach remains limited when the terminal state is not specified (e.g. when finding a projection onto manifolds).

## 6 CONCLUSION

We present a novel paradigm that integrates calculus of variations and Pontryagin's maximum principle into neural networks for learning the solutions to functional optimization problems arising in many engineering and technology and scientific problems. Our PMP-Net is unsupervised, generalizable and can be applied to general optimal control problems with moving boundaries that other related works have not addressed. We illustrate the PMP-Net strategy with two classical problems of great applied significance and show that it successfully recovers the Kalman filter and bang-bang control solutions. By leveraging the Calculus of Variations, we can analyze variations in the terminal state and time, and PMP-Net successfully optimizes this variable in the minimum time problem—something most prior works fail to do. Although these solutions have been derived analytically in the past, we experiment with these two classic problems, especially bang-bang control where no prior work has managed to use neural network to solve before, so that we can evaluate our results with the analytical optimal solutions. Our work paves the way for applying PMP-based neural networks to more complex, higher-dimensional, and analytically intractable control problems.

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

## A  CALCULUS OF VARIATION AND PONTRYAGIN'S MAXIMUM PRINCIPLE

Suppose we want to find the control $u^\star(t), t \in [0, t_f]$ that causes the system

$$\dot{x} = f(x, u)$$
$$x(0) = x_0$$

(13)

, where $f$ is a continous function with continuous partial derivatives with respect to each variable, to follow an admissible trajectory $x^\star(t), t \in [0, t_f]$ that reaches the stopping set $\mathcal{S}$, i.e., $(x(t_f), t_f) \in \mathcal{S}$ and minimizes the performance measure

$$J(x, u) = q_T(x(t_f), t_f) + \int_0^{t_f} g(x(t), u(t), t)dt$$

We consider the stopping set $\mathcal{S}$ to be of a general form $\mathcal{S} = \{(x(t), t) |\ s(x(t), t) = 0\} = \mathcal{X} \times \mathbb{R}_{\geq 0}$ where $s : \mathbb{R}^m \times \mathbb{R}_{\geq 0}$ is a differentiable function with respect to each variable. We suppose that the integrand $g$ has continuous first and second partial derivatives with respect to all of its arguments and $q_T$ has continuous first partial derivatives with respect to all of its arguments.

$$J(x, u) = q_T(x(0), 0) + \int_0^{t_f} \frac{dq_T}{dt}(x, t) + g(x(t), u(t), t)dt$$

We introduce the (vector function) Lagrange multipliers $\lambda$, also known as costates. The primary function of $\lambda$ is to enable us to make perturbations $(\delta x, \delta u)$ to an admissible trajectory $(x, u)$ while ensuring the dynamic constraints in equation 13 remain satisfied. Suppose we have an admissible trajectory $(x, u, \lambda)$ such that reaches the terminal state $x_f$ at time $t_f$, the augmented cost functional $\mathcal{L}$ is

$$\mathcal{L}(x, u, \lambda, x_f, t_f) = q_T(x_0, 0) + \int_0^{t_f} \frac{dq_T}{dt}(x, t) + g(x, u) + \lambda^T(f(x, u) - \dot{x})dt$$

$$= q_T(x_0, 0) + \int_0^{t_f} \frac{dq_T}{dt}(x(t), t) + g(x, u) + \lambda^T f(x, u) - \lambda^T \dot{x})dt$$

$$= q_T(x_0, 0) + \int_0^{t_f} \frac{dq_T}{dt}(x(t), t) + \mathcal{H}(x, u, \lambda, t) - \lambda^T \dot{x})dt$$

where the Hamiltonian $\mathcal{H} = g(x(t), u(t)) + \lambda(t)^T f(x(t), u(t))$. The calculus of variations studies how making a small pertubation to $(x, u, \lambda)$ changes the performance. Suppose the new trajectory $(x + \delta x, u + \delta u, \lambda + \delta \lambda)$ reaches new terminal state $(x_f + \delta x_f, t_f + \delta t_f)$. The change in performance

is

$$\Delta \mathcal{L} = \mathcal{L}(x + \delta x, u + \delta u, \lambda + \delta \lambda, x_f + \delta x_f, t_f + \delta t_f) - \mathcal{L}(x, u, \lambda, x_f, t_f)$$

$$= \int_0^{t_f} \frac{\partial}{\partial x} \frac{dq_T}{dt}(x, t)\delta x + \frac{\partial \mathcal{H}}{\partial x}\delta x + \frac{\partial \mathcal{H}}{\partial u}\delta u + \left(\frac{\partial \mathcal{H}}{\partial \lambda} - \dot{x}\right)^T \delta \lambda - \lambda^T \delta \dot{x} \, dt$$

$$+ \left[\frac{dq_T}{dt}(x(t_f), t_f) + \mathcal{H}(x, u, \lambda, t_f) - \lambda(t_f)^T \dot{x}(t_f)\right] \delta t_f + o(\|\delta x\|, \|\delta u\|, \|\delta \lambda\|, \|\delta t_f\|) \, dt$$

$$= \int_0^{t_f} \frac{\partial^2 q_T}{\partial x^2}\dot{x}\delta x + \frac{\partial^2 q}{\partial x \partial t}\delta x + \frac{\partial q}{\partial x}\dot{\delta x} - \lambda^T \dot{\delta x} + \frac{\partial \mathcal{H}}{\partial x}\delta x + \frac{\partial \mathcal{H}}{\partial u}\delta u + [f(x, u) - \dot{x}]^T \delta \lambda \, dt$$

$$+ \left[\frac{dq_T}{dt}(x(t_f), t_f) + \mathcal{H}(x, u, \lambda, t_f) - \lambda(t_f)^T \dot{x}(t_f)\right] \delta t_f + o(\|\delta x\|, \|\delta u\|, \|\delta \lambda\|, \|\delta t_f\|) \, dt$$

$$= \left[\frac{\partial q_T}{\partial x}\Big|_{t_f} - \lambda(t_f)^T\right]\delta x(t_f) + \int_0^{t_f} \left[\dot{\lambda} + \frac{\partial \mathcal{H}}{\partial x}\right]\delta x + \frac{\partial \mathcal{H}}{\partial u}\delta u + [f(x, u) - \dot{x}]^T \delta \lambda \, dt$$

$$+ \left[\frac{\partial q_T}{\partial x}\Big|_{t_f}\dot{x}(t_f) + \frac{\partial q_T}{\partial t}\Big|_{t_f} + \mathcal{H}(x, u, \lambda, t_f) - \lambda(t_f)^T \dot{x}(t_f)\right]\delta t_f + o(\|\delta x\|, \|\delta u\|, \|\delta \lambda\|, \|\delta t_f\|) \, dt$$

$$= \left[\frac{\partial q_T}{\partial x}\Big|_{t_f} - \lambda(t_f)^T\right]\delta x_f + \int_0^{t_f} \left[\dot{\lambda} + \frac{\partial \mathcal{H}}{\partial x}\right]\delta x + \frac{\partial \mathcal{H}}{\partial u}\delta u + [f(x, u) - \dot{x}]^T \delta \lambda \, dt$$

$$+ \left[\frac{\partial q_T}{\partial t}\Big|_{t_f} + \mathcal{H}(x, u, \lambda, t_f)\right]\delta t_f + o(\|\delta x\|, \|\delta u\|, \|\delta \lambda\|, \|\delta t_f\|) \, dt$$

The fundamental theorem of calculus of variation states that if $(x^\star, u^\star)$ is extrema, then the variations $\delta J$ (linear terms of $\delta x, \delta u, \delta x_f, \delta t_f$) must be zero. Since $\lambda$ can be chosen arbitrarily, we choose $\lambda^\star$ such that the linear terms of $\delta x$ is 0, i.e.

$$\dot{\lambda}^\star + \frac{\partial \mathcal{H}}{\partial x}\Big|_\star = 0 \tag{14}$$

Since the $(x^\star, u^\star)$ must satisfy the constraint in equation 13,

$$f(x^\star, u^\star) - \dot{x}^\star = 0 \tag{15}$$

The remaining variation $\delta u$ is independent, so its coefficient must be zero; thus,

$$\frac{\partial \mathcal{H}}{\partial u}\Big|_\star = 0 \tag{16}$$

The rest of variations are therefore 0, i.e.,

$$\left[\frac{\partial q_T}{\partial x}\Big|_{\star, t_f^\star} - \lambda(t_f^\star)^T\right]\delta x_f + \left[\frac{\partial q_T}{\partial t}\Big|_{\star, t_f^\star} + \mathcal{H}(x^\star, u^\star, \lambda^\star, t_f^\star)\right]\delta t_f = 0 \tag{17}$$

We consider two special cases that present in our experiment: 1) the terminal state $x_f$ is fixed, and 2) the terminal time is fixed.

1) First, if the terminal state is fixed and terminal time is free, i.e., $\delta x_f = 0$. Then $\delta t_f$ can be arbitrary and coefficients of $\delta t_f$ must be 0, i.e.,

$$\left[\frac{\partial q_T}{\partial t}\Big|_{\star, t_f^\star} + \mathcal{H}(x^\star, u^\star, \lambda^\star, t_f^\star)\right] = 0$$
$$x^\star(t_f^\star) = x_f \tag{18}$$

2) Now we consider the case where the terminal time is fixed and the terminal state is free, i.e., $\delta t_f = 0$. Then $\delta x_f$ can be arbitrary and coefficients of $\delta x_f$ must be 0, i.e.,

$$\left[\frac{\partial q_T}{\partial x}\Big|_{\star, t_f^\star} - \lambda(t_f^\star)^T\right] = 0$$
$$t_f^\star = t_f \tag{19}$$

But in more general cases where $\delta x_f, \delta t_f$ are related, the equation 17 reduces to (Athans & Falb, 1996)

$$
\frac{\partial q_T}{\partial x}(x^\star(t_f), t_f) - \lambda^\star(t_f) = \sum_{i=1}^{k} d_i \left[ \frac{\partial s_i}{\partial x}(x^\star(t_f), t_f) \right] \quad (\mathcal{B}_1)
$$

$$
\mathcal{H}(x^\star(t_f), u^\star(t_f), \lambda^\star(t_f)) + \frac{\partial q_T}{\partial t}(x^\star(t_f), t_f) = \sum_{i=1}^{k} d_i \left[ \frac{\partial s_i}{\partial t}(x^\star(t_f), t_f) \right] \quad (\mathcal{B}_2)
$$

(20)

The equation 18 and equation 19 allow us to determine the optimal terminal state or optimal terminal time when they are free and to be optimized. These equations from 14 to 19 are called Pontryagin's maximum principle.

## B  KALMAN FILTERING DERIVATION

The performance measure for designing optimal linear filter is

$$
J(\Sigma, G) = q_T(\Sigma(T), T)
$$
$$
= \operatorname{tr}(\Sigma(T))
$$

Since the terminal time $t_f = T$ is specified and terminal state is free, equation 19 applies. Pontryagin's maximum principle yields

$$
\dot{\Sigma}^\star = \left[ A - G^\star C \right] \Sigma + \Sigma \left[ A - G^\star C \right]^T + BQB^T + G^\star R G^{\star T} \tag{21a}
$$

$$
\left. \frac{\partial \operatorname{tr}(\lambda^\star \dot{\Sigma}^\star)}{\partial \Sigma} \right|_\star + \dot{\lambda^\star}^T = 0 \tag{21b}
$$

$$
\left. \frac{\partial \operatorname{tr}(\lambda^\star \dot{\Sigma}^\star)}{\partial G} \right|_\star = 0 \tag{21c}
$$

$$
\lambda^\star(T)^T = \boldsymbol{I}_n \tag{21d}
$$

$$
\Sigma^\star(0) = \Sigma_0 \tag{21e}
$$

Simplifying equation 21b yields,

$$
\dot{\lambda}^\star = -\lambda^\star \left[ A - G^\star C \right] - \left[ A - G^\star C \right]^T \lambda^\star \tag{22}
$$

From equation 21d and equation 22, we can conclude that $\lambda^\star$ is symmetric positive definite. Substitute $\dot{\Sigma}^\star$ in equation 21c by R.H.S expression in equation 21a yields,

$$
2\lambda^\star \left[ 2G^\star R - 2\Sigma^\star C^T \right] = 0 \tag{23}
$$

Since $\lambda^\star$ is invertible,

$$
G^\star = \Sigma^\star C^T R^{-1} \tag{24}
$$

Plugging this solution $G^\star$ in equation 21a yields

$$
\dot{\Sigma}^\star = A\Sigma^\star + \Sigma^\star A^T + BQB - \Sigma^\star C^T R^{-1} C \Sigma^\star \tag{25}
$$

which is the matrix differential equation of the Riccati type. The solution $\Sigma^\star$ can be derived from the initial condition $\Sigma^\star(0) = \Sigma_0$ and the differential equation 25.

## C  BANG-BANG CONTROL DERIVATION

Since the terminal state $x_f$ is specified and terminal time is free, equation 18 applies. Pontryagin's maximum principle yields

$$
\dot{x}^\star = \begin{bmatrix} x_2^\star \\ 0 \end{bmatrix} + \begin{bmatrix} 0 \\ u^\star \end{bmatrix} \tag{26a}
$$

$$\dot{\lambda}^{\star} = \begin{bmatrix} 0 \\ -\lambda_1^{\star} \end{bmatrix} \tag{26b}$$

$$u^{\star} = \arg\min_{u} 1 + \lambda_1^{\star} x_2^{\star} + \lambda_2^{\star} u \tag{26c}$$

$$1 + \lambda_1^{\star}(t_f^{\star}) x_2^{\star}(t_f^{\star}) + \lambda_2^{\star}(t_f^{\star}) u^{\star}(t_f^{\star}) = 0 \tag{26d}$$

$$\begin{bmatrix} x_1^{\star}(0) \\ x_2^{\star}(0) \end{bmatrix} = \begin{bmatrix} x_0 \\ v_0 \end{bmatrix} \tag{26e}$$

$$\begin{bmatrix} x_1^{\star}(t_f^{\star}) \\ x_2^{\star}(t_f^{\star}) \end{bmatrix} = \begin{bmatrix} 0 \\ 0 \end{bmatrix} \tag{26f}$$

The equation 26c yield

$$\forall u, 1 + \lambda_1^{\star} x_2 + \lambda_2^{\star} u^{\star} \leq 1 + \lambda_1^{\star} x_2 + \lambda_2^{\star} u$$

$$u^{\star} = \begin{cases} -\text{sign}(\lambda_2) & \text{if } \lambda_2^{\star} \neq 0 \\ \text{indeterminate} & \text{if } \lambda_2^{\star} = 0 \end{cases}$$

Assuming that $\lambda_2^{\star}$ is **not** a zero function, the equation 26b yields

$$\begin{aligned} \lambda_1^{\star}(t) &= c_1 \\ \lambda_2^{\star}(t) &= -c_1 t + c_2 \end{aligned} \tag{27}$$

where $c_1, c_2$ are constants to be determined. We see from equation 27 that $\lambda_2$ changes sign at most once. There are two possible cases:

1. $\lambda_2^{\star}$ sign remains constant in $[0, t_f^{\star}]$

2. $\lambda_2^{\star}$ changes sign in $[0, t_f^{\star}]$

For case 1, we have the general form of

$$\begin{aligned} x_2(t) &= v_0 + at && \text{for } t \in [0, t_f^{\star}] \\ x_1(t) &= p_0 + v_0 t + \frac{1}{2} a t^2 && \text{for } t \in [0, t_f^{\star}] \end{aligned} \tag{28}$$

For case 2, we have the general form of $x$

$$x_2(t) = \begin{cases} v_0 + at & \text{if } t \leq t_m \\ v_0 + a t_m - a(t - t_m) & \text{if } t_f^{\star} \geq t \geq t_m \end{cases}$$

$$x_1(t) = \begin{cases} p_0 + v_0 t + \frac{1}{2} a t^2 & \text{if } t \leq t_m \\ x_0 + v_0 t + 3 a t t_m - 2 a t_m^2 - \frac{1}{2} a t^2 & \text{if } t \geq t_m \end{cases} \tag{29}$$

where $a = \pm 1$ and $t_m$ is the time where $\lambda_2^{\star}$ switches sign. To determine which case corresponds to the system, we validate with the boundary condition. Suppose we try with the general expression in equation 29 and substitute in boundary conditions in equation 26:

$$\begin{aligned} -c_1 t_m + c_2 &= 0 && (\text{ From the condition } \lambda_2^{\star}(t_m) = 0) \\ a &= \pm 1 \\ v_0 + a t_m - a(t_f - t_m) &= 0 && (\text{ From equation 26f}) \\ p_0 + v_0 t + 3 a t_f t_m - 2 a t_m^2 - \frac{1}{2} a t_f^2 &= 0 && (\text{ From equation 26f}) \\ 1 - a(c_1 t_f + c_2) &= 0 && (\text{ From equation 26d}) \end{aligned} \tag{30}$$

Specific example: $x_0 = 1, v_0 = 0$.

Solving equation 30 yields

$$
\begin{aligned}
t_f &= 2t_m \\
1 &= -a t_m^2 \\
a &= -1 \\
t_m &= 1 \\
c_1 &= 1 \\
c_2 &= 1
\end{aligned}
\tag{31}
$$

which means the system with initial state condition $(1, 0)$ falls the second case. If we substitute the general expression equation 28 instead, there would be no solutions satisfying equation 30.

**Remarks**: This derivation of bang-bang solution is based on assumption that $\lambda$ is not a zero function.

