# OpenReview forum: "Is Pontryagin's Maximum Principle All You Need? Solving optimal control problems with PMP-inspired neural networks"
_ICLR.cc/2025/Conference — Submitted to ICLR 2025_

### Official Review · Reviewer_jkng · 2024-10-29

**Soundness:** 3
**Presentation:** 3
**Contribution:** 2
**Rating:** 5
**Confidence:** 4

**Summary:**

In this paper, the authors integrate Pontryagin's Maximum Principle (PMP) into a neural network architecture, termed PMP-Net, to address optimal control problems. Unlike most existing methods, PMP-Net learns optimal control policies in an unsupervised manner, which represents the paper's main contribution. The authors evaluate PMP-Net on classical control tasks, including the Kalman filter and the bang-bang control problem, demonstrating satisfying performance compared existing approaches.

**Strengths:**

* The concept of integrating Pontryagin's Maximum Principle (PMP) into neural networks to solve optimal control problems is interesting, as optimal control is crucial across various domains, and PMP serves as a foundational method for many existing solution approaches.
* A notable advantage of PMP-Net is its ability to be trained without labeled data, addressing the high costs typically associated with data labeling in control problems.

**Weaknesses:**

* While the motivation for PMP-Net is stated clearly, the full benefits of this approach are not adequately demonstrated in the paper.
* The experimental evaluation lacks depth, as it does not compare PMP-Net with other relevant methods, such as Physics-Informed Neural Networks (PINNs) and Hamiltonian Neural Networks (HNNs), which were introduced in the introduction but not used for comparative analysis.
* Since PMP shares similarities with the well-known Karush-Kuhn-Tucker (KKT) conditions, and many works have explored integrating KKT into neural networks (e.g., OptNet, DC3), it would be helpful for the authors to differentiate PMP-Net from existing KKT-based differentiable solvers or neural networks.
* There is a concern regarding constraint satisfaction in Equation (4). When (x, u) satisfies the dynamic constraint \dot{x} = f(x,u), the L(x, u, \lambda) = f(x, u) is just a necessary but insufficient condition. In section 2.2, the authors state as "In our following experiments in Section 3 and 4, ..., PMP-net to enforce hard constraints and to allow PMP-net to learn when terminal time in unknown." However, I don't find effective approaches which can "enforce" hard constraints in these two sections.
* The experimental design is quite basic, limiting PMP-Net’s relevance to real-world optimal control problems. Additionally, the experiments lack comparisons on solution time, which would be relevant for practical applications.

**Questions:**

* What differentiates contributions 1 and 2? It seems that incorporating PMP into the training methodology would inherently include handling dynamic constraints and other variable-related constraints.
* What does the term "learning paradigms" in contribution 3 mean? Sections 3 and 4 do not appear to outline any specific paradigms for PMP-Net.
* Why does the baseline diverge in the Kalman filtering experiment? Additional detail would clarify, as neural network controllers often face divergence issues.
* Why are baseline methods not compared in the bang-bang control experiment?

---

### Official Review · Reviewer_m11v · 2024-11-03

**Soundness:** 3
**Presentation:** 3
**Contribution:** 3
**Rating:** 5
**Confidence:** 3

**Summary:**

In domains, such as telecommunications, automation, robotics, and control systems, labelled data in these domains are often scarce, difficult, or expensive to be obtained. Providing prior knowledge, specifically existing design principles that have been successful in various engineering, scientific, and technology practices. This article first uses Pontryagin’s Maximum Principle as a soft constraint to incorporate prior knowledge, and designs dynamical constraints and other constraints on the variables and functions of interest. This method mimics the design of feedback
controllers used in optimal control. Additionally, it also replicates the design of the Kalman filter and the bang-bang control
without using labelled data.

**Strengths:**

1. Incorporate Pontryagin’s Maximum Principle as soft constraints in ML training methodology.
2. Involving the design for dynamical constraints and other constraints on the variables and functions of interest
3. Propose learning paradigms that effectively train PMP-net to derive the optimal solution.
4. PMP-net replicates the design of the Kalman filter and the bang-bang control without using labelled data.

**Weaknesses:**

1. There is no systematic explanation of some ambiguity. For example, in Figure 2 (a), why can PMP-Net with curriculum training reach convergence faster than PMP-Net with standard training?
2. The visualisation result is not clear. Figures should be revised again. For example, the colour of lines and titles of the x and y axes.
3. More experiments are required, such as some applications of the Kalman filter and PMP-Net on real data.

**Questions:**

1. In Figure 2 (c), PMP-Net $G_{11}$ looks not convergent at the end. Can you explain why and how this phenomenon can affect the stability?
2. What about the real application performance on filtering or controlling tasks?
3. How to set the time period $t$? Manually or empirically?
4. What about the gradient vanishing or exploding problem when using PMP-Net?

---

### Official Review · Reviewer_pDZu · 2024-11-07

**Soundness:** 3
**Presentation:** 3
**Contribution:** 1
**Rating:** 3
**Confidence:** 5

**Summary:**

This paper proposes Pontryagin’s Maximum Principle Neural Network (PMPnet) to solve the optimal control problem. The problem is defined in equation 3, and using calculus of variation, the necessary conditions for optimality are given in equation 5, known as Pontryagin’s maximum principle, so is the name of the network. Such equations, instead of conventional methods like shooting, etc, are solved using neural networks (see Fig 1 or 3 for instance) by including the equations as soft constraint and using inductive bias to design the architecture. They solve two canonical problems optimal filtering and bang bang control with unknown time interval and recover the known solutions in an unsupervised fashion, to demonstrate the applicability of PMPnet.

**Strengths:**

The paper is well-written.
The engineered neural network for PMP-net is novel and interesting.
The free final time is an interesting problem that authors solve. It's a nice verification for both examples that have known solutions (Kalman filter, and bang-bang control).

**Weaknesses:**

There are works where optimal control formulation is implmented using soft constraints.
Also solution of necessary conditions of optimality with PINNs is not new, e.g. see ONN: An adjoint-oriented neural network method for all-at-once solutions of parametric optimal control problems

In fact, there is somehwat similar work by D’Ambrosio et al back in 2021, Pontryagin Neural Networks with Functional Interpolation for Optimal Intercept Problems, and in turn this has been used for several examples, e.g in quantum optimal control of An Application of Pontryagin Neural Networks to Solve Optimal Quantum Control Problems

Now approaches of above papers and this work are not the same; in fact the archotecture of Fig 1 can be considered as a novel work but this alone does not bring enough insight or contribution for the paper to be considered for publication in ICLR. This is become more pronounced considering the two examples that are really not as involved as the examples solved in above papers; so the case studies also do not manifest something unique about this approach that has not been addressed in the past.

In summary, solving necessary conditions of optimality and/or Pontryagin's maximum principle as a soft constraint in the context of PINNs has been proposed and used to solved more complicated examples in the literature. A slightly novel acrchitecture proposed in this paper does not add a meaningful contribution to the community.

**Questions:**

Between Figs 1 and 3, do we need to propose a different architecture for each optimal control? They look essentially similar, so maybe no need to repeat the figure (of course the optimality and feasibility equations are different for each case).

It's very essential to expand related work section for various works where PINNs are used to solve optimal control and position this work in reference to such body of literature and draw conclusions about strengths of this work

The solution of this problem is for a given set of paramters. So PMP-net is for an "instance". Any comments how this work can be applied to cases where there are uncertainties (unknown coefficients in the equations)?

What if instead of PMP-net, euthos simply solve a NN with a loss function that satisfies the dynamics as well as the performance metric i.e. equation 3. In fact Mowlavi and Nabi Optimal control of PDEs using physics-informed neural networks solved the optimal control this way. What would be the advantage of PMP-net? On one hand you are enforcing optimality but on the other hand you are making the landscape of optimizaiton more complicated.

---

### Official Review · Reviewer_8pyS · 2024-11-08

**Soundness:** 2
**Presentation:** 1
**Contribution:** 2
**Rating:** 5
**Confidence:** 4

**Summary:**

The paper proposes a method to estimate the state and costate for optimal control problems that align with the optimality conditions expressed by Pontryagin’s Maximum Principle (i.e., the generalization of the Euler-Lagrange equation within the context of optimal control). The core contribution of the manuscript is to provide an algorithm that translates the stationarity conditions of the constrained optimization problem into a loss function for the parameters of the networks that estimate the state and costate. The paper then demonstrates that, in two relevant control problems, optimization of such losses recovers (at least asymptotically) the known optimal value.

**Strengths:**

The approach is interesting and highlights a natural connection between learning with dynamical constraints and optimal control.

**Weaknesses:**

See Question section below.

**Questions:**

While the overall approach is interesting and quite natural, I have several questions, doubts, and comments that I will now try to express as precisely as I can.

1. While the method is proposed as an unsupervised learning problem, it is not clear to me where the learning lies. To be more explicit, in reading the manuscript, although terms like “learning” and “training” are used, I only see a gradient-based solution to an optimization problem involving parametric functions. As far as I can understand, there is no generalization process involved, no “ambition” that the optimized weights of the network could effectively solve even a slightly different problem (such as the same optimal control on a different temporal horizon).

2. A related comment/question to point 1 is the following: If I’ve understood correctly, the networks that estimate the costate and the state take the temporal variable as input. This itself suggests that no learning is actually involved; it would be like trying to solve a (for instance, supervised) learning problem on sequences by merely mapping the sequence indices (1,2,3,\dots) to the target. Again, it seems to me that this approach is simply an optimization method on the weights of a neural network to approximate a solution to an optimal control problem. I might be missing something, so it would be helpful to receive clarification from the authors on this point.

3. I don’t understand the expression of the losses around lines 241–244. In particular, why are we using  $x$  instead of  $\Sigma$  as in Eq. (9)? I might be missing something important here. Why does the loss term that contains the ODE for the costate impose that  $\dot{\lambda} = - \mathcal{H}$  instead of  $\dot{\lambda} = - \mathcal{H}_\Sigma$  (where the subscript here denotes a partial derivative)?

4. Similarly, I don’t understand the remark on lines 250–254: “For fair evaluation…” Isn’t the state in this example the covariance matrix  $\Sigma$?

5. I would like a more detailed explanation of why the solutions found only approximate the asymptotic value of the optimal control solution. Why is this the case? Why do you think your approach yields such a result?

6. In your review of the literature, you overlooked a line of research that is extremely close to what you are proposing, primarily conducted by a research group that used Calculus of Variations to define problems in lifelong learning. See for instance

- Betti, Alessandro, and Marco Gori. "The principle of least cognitive action." Theoretical Computer Science 633 (2016): 83-99.

- Betti, Alessandro, Marco Gori, and Stefano Melacci. "Cognitive action laws: The case of visual features." IEEE transactions on neural networks and learning systems 31.3 (2019): 938-949.

- Tiezzi, Matteo, et al. "Focus of attention improves information transfer in visual features." Advances in Neural Information Processing Systems 33 (2020): 22194-22204.

More recently, they also proposed approaches to learning that utilize optimal control. Similar to your approach, they propose training a neural network to estimate the costate:

- Betti, Alessandro, et al. "Neural Time-Reversed Generalized Riccati Equation." Proceedings of the AAAI Conference on Artificial Intelligence. Vol. 38. No. 8. 2024.

7. The paper uses non-standard definitions of the Lagrangian and Hamiltonian, which makes it more difficult to read than necessary. See, for instance:

- Giaquinta, Mariano, and Stefan Hildebrandt. Calculus of variations II. Vol. 311. Springer Science & Business Media, 2013.

- Cannarsa, Piermarco, and Carlo Sinestrari. Semiconcave functions, Hamilton-Jacobi equations, and optimal control. Vol. 58. Springer Science & Business Media, 2004.

- Evans, Lawrence C. Partial differential equations. Vol. 19. American Mathematical Society, 2022.

- Bardi, Martino, and Italo Capuzzo Dolcetta. Optimal control and viscosity solutions of Hamilton-Jacobi-Bellman equations. Vol. 12. Boston: Birkhäuser, 1997.

8. The expression of the PMP is incomplete, and the derivation in Appendix A relies on several unstated assumptions. In which functional space is the optimization problem defined? In which space do the variations lie (this is also crucial for giving meaning to the boundary conditions)? What are the regularity assumptions on  $f$  and on the Hamiltonian? For instance, Eq. (9) does not make sense if the Hamiltonian is not  $C^1$ , and it is often beneficial to assume the Hamiltonian to be  $C^{1,1}_{\text{loc}}$.

I would like to understand better about the point that I raise
and possibly increase my score.

---

### Author Response · Authors · 2024-11-28
**Modifications in the revised paper**

We would like to thank all the reviewers for their constructive feedback and evaluations. We took all comments and suggestions from reviewers and revised our paper.

Here are the main modifications in our revised paper:
1. The Revised related work section addresses more clearly how our work is different from existing work using PINNs to solve optimal control
2. The experiment in the inference section is slightly modified. We add one more baseline to the experiment and fix our results.
3. Section 2 defines a more general optimal control problem than what other previous work did, i.e., the terminal state and time are in the stopping set.

We would appreciate it if everyone read the whole revised paper again and re-evaluate. In the meantime, we will be addressing all the questions by reviewers.

---

### Meta-Review · Area_Chair_y43Z · 2024-12-17

**Metareview:**

This paper introduces Pontryagin’s Maximum Principle Neural Network (PMPNet) to solve optimal control problems. The key idea is to leverage Pontryagin’s Maximum Principle (PMP)—necessary optimality conditions—to design both the network architecture and the training loss function, incorporating the PMP equations as soft constraints. The method is demonstrated on two canonical problems: optimal linear filtering and bang-bang control with unknown time intervals.

The paper is overall well-organized and clearly written. A notable strength of PMPNet, compared to other learning-based approaches that optimize a performance metric, is its applicability to control problems where the performance metric cannot always be computed, such as minimum-time control problems. However, a major limitation is the omission of classical numerical methods traditionally used for such problems, particularly those solving the two-point boundary value problems derived from PMP. Established software packages, such as bvp4c in Matlab or solve_bvp in SciPy, are highly mature and efficient for this purpose. The paper does not discuss how PMPNet compares with these methods in terms of accuracy, computational efficiency, or ease of implementation.

Furthermore, as noted by multiple reviewers, the idea of using optimality conditions as soft constraints for training neural networks is not novel. Combined with the lack of strong numerical evidence demonstrating a clear advantage of PMPNet over existing classical or learning-based approaches, the paper has not met the bar for publication at ICLR. I encourage the authors to address these issues, as the work has the potential to make a meaningful contribution with further improvement.

**Additional Comments On Reviewer Discussion:**

During the discussion period, the authors clarified a few conceptual questions and updated the manuscript to emphasize the general formulation of the control problem, which they argue cannot be addressed by existing learning-based methods like PINNs. However, the work still lacks a comparison with classical numerical methods. Furthermore, in the Kalman filtering example, where learning methods based on optimizing performance metrics are applicable, it remains unclear whether PMPNet offers any tangible advantage. This concern was reiterated by Reviewer m11v, who questioned the real-world performance of PMPNet.

The authors argued that “PMPNet learns the optimal solution that an expert control researcher would derive analytically with significant effort by applying the PMP conditions.” However, this argument still overlooks the existence of mature numerical methods capable of solving PMP equations efficiently.

---

### Decision · Program_Chairs · 2025-01-22

Reject